# Product innovation design process combined Kano and TRIZ with AD: Case study

Hui Rong[1,2], Wei Liu[1,2]*, Jin Li[1,2], Ziqian Zhou[1,2]

**1** College of Furnishing and Industrial Design, Nanjing Forestry University, Nanjing, Jiangsu, China, **2** Co-Innovation Center of Efficient Processing and Utilization of Forest Resources, Nanjing Forestry University, Nanjing, Jiangsu, China

☯ These authors contributed equally to this work.
\* liuwei@njfu.edu.cn

**Data Availability Statement:** All relevant data are within the paper and its Supporting Information files.

## Abstract

In the era of rapid product iteration, companies need simple and effective methods to guide the entire process of product innovation design and enhance their product innovation capabilities. Most research focused on improving one or several steps in the product design process. Although some scholars have proposed methods that guided the entire process, they combined more than three different theories, which increased the difficulty of theoretical learning and the complexity of practical implementation. This paper proposed a product innovation design process composed of three theoretical methods: Kano, Axiomatic Design (AD), and Theory of the Solution of Inventive Problems (TRIZ). This new process guided the entire product design process with fewer theoretical methods, reducing the difficulty of learning and implementation. The paper demonstrated the effectiveness of this method through the design practice of a portable two-wheeled self-balancing vehicle. Additionally, the discussion section explored the method's potential from the design management perspective.

## Introduction

In the context of fast-paced product iteration, the design innovation capability of companies needs to be enhanced [1]. So, the design innovation methods and processes must be effective and as simple as possible. To propose practical methods, it is necessary to outline the basic design innovation process. Improving user satisfaction is the ultimate goal of continuous product innovation [2]. Therefore, better services and products must start by analyzing user needs [3]. User needs determine the product's functional attributes, and implementing these functions relies on technical principles and product structure [4]. Analyzing the mapping relationship between functions and structures and resolving conflicts are the main tasks of product functional design [5]. After completing the design solution, evaluating whether the product has been improved and optimized is necessary, which will help designers make design decisions [6]. In summary, the complete process of product innovation design includes five main steps: user needs research (S1), requirement-function transformation (S2), analysis of design issues (S3), resolution of design issues (S4), and design solution evaluation (S5).

**Funding:** The National Natural Science Foundation of China (NOS:52105262)". Professor Wei Liu was the sponsor of the paper research. She played the roles of project management and writing guidance in this study. The authors did not receive any salary from the funder.

**Competing interests:** The authors have declared that no competing interests exist.

Previous studies have proposed methods that only cover a single or a few steps in the product design process. These methods cannot guide the entire process of product innovation design. Further literature research has revealed that some scholars have paid attention to the need for strategies that can guide the whole process. However, these methods required the integration of more than three theories, which brought forth other issues. Readers needed a broad theoretical foundation to comprehend research articles and the design process. Additionally, the abundance of methods increased the complexity of the design practice, making it challenging for readers to learn and difficult to apply these methods widely.

The paper proposed an integrated method that covered the entire design process, utilizing only three theoretical approaches. This new method ensures guidance throughout the design workflow while reducing the complexity of both theory and practice. The proposed approach involves the use of Kano analysis for understanding user requirements (S1), utilizing the AD theory for requirement-function transformation, design problem analysis, and design solution evaluation (S2, S3, S5), and incorporating TRIZ for design problem-solving (S4). The paper demonstrated the new process's effectiveness through a design practice.

## Literature review

This literature review introduces the basic theories and research applications of Kano, AD, and TRIZ methods, their advantages and disadvantages, and the research gap.

### Kano

Kano, proposed by Professor Noriaki Kano, is a two-dimensional cognitive model that studies the nonlinear relationship between product quality performance and user satisfaction [7]. The Kano model is widely applied in product development to help designers clarify user needs and establish product objectives [8]. The theoretical model classifies user needs into five categories based on the relationship between the completeness of functional requirements and user satisfaction: Must-Requirement (M), One-Dimensional Requirement (O), Attractive Requirement (A), Indifferent Requirement (I), and Reverse Requirements (R) [9–12]. These classification criteria assist in identifying explicit and implicit user needs, which contribute to designing new products or improving existing ones [13]. AD theory lacks specific methods for discovering and analyzing user needs. Kano can compensate for AD's shortcomings and enhance the accuracy of mapping user needs to functional requirements.

### Axiomatic design

Professor Suh proposed Axiomatic Design (AD) in 1976 [14, 15]. This theoretical approach aims to establish design specifications quantitatively, assisting designers in enhancing the logic and systematics of problem analysis [16]. The theory proposes the concepts of customer domain (CAs), functional domain (FRs), structural domain (DPs), and process domain (PVs) to standardize the design process [17]. A mapping relationship exists between these domains in a zigzag pattern, as shown in Fig 1. AD utilizes design matrices to express the relationships between the functional and structural domains and employs the independence axioms to analyze design problems [18, 19]. When there is a coupled design, it indicates a contradiction between function and structure, and designers need to find ways to decouple them [20, 21]. After obtaining design solutions, the information axioms in AD help designers make decisions by quantitatively comparing the information content in design alternatives [22]. Therefore, AD can assist in analyzing problems and evaluating solutions. However, previous research has indicated two notable limitations of AD. Firstly, although AD introduces the concept of the customer domain, it fails to provide an accurate method for identifying user needs and

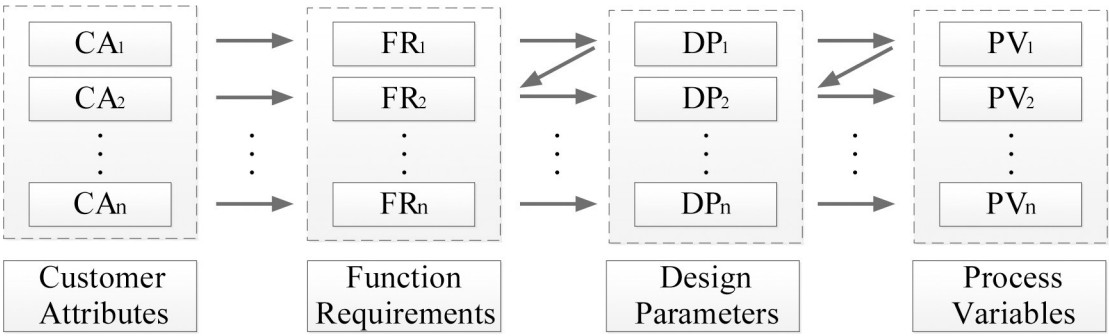

**Fig 1. The Z-shaped mapping relationship between the domains.**

classifying their priorities [23]. Secondly, the ability to provide appropriate design suggestions is insufficient when presenting concrete solutions [24]. These are the main reasons the paper proposes the integration of Kano, TRIZ, and AD.

## Theory of inventive problem solving

Theory of Inventive Problem Solving(TRIZ) has been widely and rapidly adopted in academic and industrial fields as an efficient method for solving conflicting problems [25]. In 1946, Altshuller proposed TRIZ based on the analysis of interdisciplinary invention patents [26]. Since then, Ilevbare et al. have perfected the theoretical system of TRIZ by summarizing the contradiction matrix, 40 principles of the invention, and other related theoretical models, tools, and research methods [27]. TRIZ summarizes innovation methods from all fields and proposes a universal pattern that can be applied to engineering, tourism, catering, sharing economy, e-commerce, and other areas [28]. By addressing problems at the system level, TRIZ can effectively balance different interests in product design [29], and many designers and product design scholars have achieved effective design results using TRIZ [30, 31].

The researcher classified all the tools and methods in the TRIZ theory system into three levels: philosophy, principles, and tools [32, 33], as shown in Fig 2. In actual use, researchers can flexibly choose the required methods based on the specific situation. Using contradictions to define problems and solve them with a contradiction matrix is one of the most commonly used methods in TRIZ theory [34]. TRIZ theory describes two types of contradictions: technical and physical contradictions. When a technical contradiction arises, changing one attribute

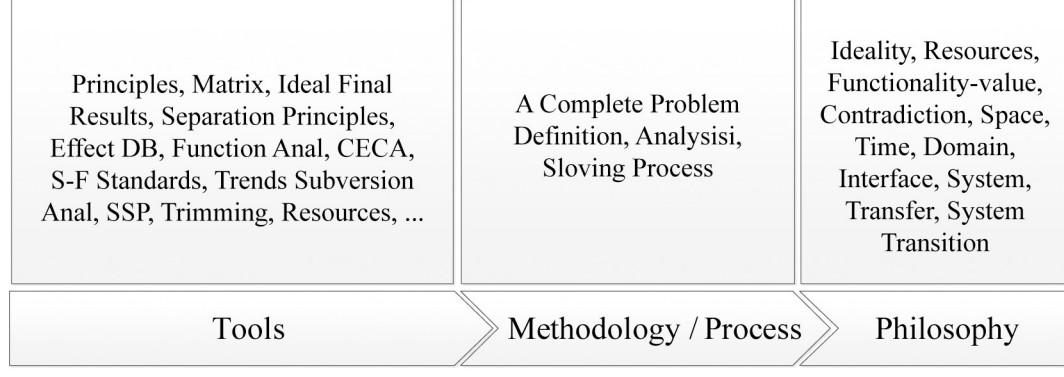

**Fig 2. Summary and classification of TRIZ theory.**

in the system will lead to the deterioration of others. When a physical contradiction arises, the same parameter in the system cannot exist under different requirements [35]. Inventive principles will be found by analyzing the contradictions and comparing them to the corresponding engineering parameters and contradiction matrix [36]. Many published studies have demonstrated the effectiveness of this method [37–39]. For designers, the methods in TRIZ theory can provide new design inspiration and help solve contradictions that arise during the design process [40, 41]. After analyzing design problems using AD, designers can utilize TRIZ to help solve these problems, which will compensate for the lack of solution methods in AD.

## Research gap

This study presents the literature through tables to provide a detailed overview of the existing research status. Due to the extensive literature volume, the tables only include research papers that meet the following two criteria: i. The papers were published within the past five years; ii. The papers proposed methodological processes that covered at least two steps in the design process. Table 1 displays the correspondence between the full names and abbreviations of theoretical methods, while Table 2 presents the relevant references.

Numerous scholars have proposed improved methods for individual or multiple steps in product innovation design. Reference [42] examined the current needs of users using Kano (S1). References [43, 44] utilized mathematical models and algorithms to predict future product demands (S1). Reference [45] established the connection between user requirements and product functions by QFD (S2). Additionally, it introduced a robot concept design for architectural layouts through TRIZ (S4). References [46–48] analyzed contradictions between functionalities and structures with AD (S3) and subsequently proposed design solutions by applying TRIZ (S4). Reference [50] constructed an evaluation system through literature research and the entropy evaluation method and evaluated mobile applications in government services by grey relational analysis (S5). Reference [52] employed qualitative methods to determine user requirements for bicycle handlebars (S1). Then, it proposed solution strategies using TRIZ (S4) and evaluated user satisfaction with improved handlebars by utilizing Kano and IPA (S5). These methodological processes presented in the studies can effectively enhance the efficiency or accuracy of specific steps in the design process, but they did not provide guidance for the entire process of product innovation design.

Further literature research reveals that some scholars have noticed the research gap in existing methods that failed to cover the entire innovation design process. To address this issue,

**Table 1. The full names and abbreviations of the theoretical methods.**

| Full Name | Abbreviation | Full Name | Abbreviation |
|---|---|---|---|
| Axiomatic Design | AD | Important-Performance Analysis | IPA |
| Analytic Hierarchy Process | AHP | Kano | Kano |
| Algorithm for Inventive-Problem Solving | ARIZ | KJ Method | KJ |
| Design Structure Matrix | DSM | Life Cycle Evaluation | LCE |
| Decision Making Trial and Evaluation Laboratory | DEMATEL | Principal Component Analysis | PCA |
| Evaluation Grid Method | EGM | PUGH Concept Selection Matrix | PCM |
| Function Analysis System Technique | FAST | Quantification Theory Type I | QTT1 |
| Failure Mode and Effects Analysis | FMEA | Quality Function Deployment | QFD |
| Finite Element Analysis | FEA | Quality Function Deployment Combined with Environmental Aspects | QFDE |
| Grey Relational Analysis | GRA | Semantic Difference | SD |
| House of Quality | HOQ | Technique for Order Preference by Similarity to an Ideal Solution | TOPSIS |
| Interpretive Structural Modeling | ISM | Theory of Inventive Problem Solving | TRIZ |

**Table 2. The references on research related to product innovation design.**

| Number of Methods | Reference | Tool used | S1-S5 | | | | | Brief Description |
|---|---|---|---|---|---|---|---|---|
| | | | S1 | S2 | S3 | S4 | S5 | |
| Two | [45] | QFD/TRIZ | | √ | √ | √ | | A conceptual robot design for an automated layout of building structures was proposed using QFD and TRIZ. |
| | [46] | AD/TRIZ | | | √ | √ | | The paper designed a cutter-changing robot using modular design, incorporating AD and TRIZ. |
| | [47] | AD/TRIZ | | | √ | √ | | The paper helped designers utilize AD and TRIZ under incomplete information environments. |
| | [48] | AD/TRIZ | | | √ | √ | | In modular design, a new design approach for module interfaces was proposed by AD and TRIZ. |
| | [49] | Kano/QFD | √ | √ | √ | | | The paper utilized Kano and QFD to identify user requirements and design elements, proposing an optimized design for wooden office desks. |
| Three | [51] | Kano/QFDE/TRIZ | √ | √ | | √ | | The paper utilized Kano and QFD to identify requirements and proposed product structures by TRIZ. |
| | [52] | TRIZ/Kano/IPA | √ | | | √ | √ | The paper qualitatively determined user requirements and employed TRIZ to conduct an improvement design of bicycle handlebars. Kano and IPA evaluated the new solution. |
| | [53] | AHP/QFD/PCM | √ | √ | √ | | √ | The paper utilized AHP and QFD to analyze the needs of surgical healthcare professionals and the design characteristics of medical devices. PCM evaluated the design proposals. |
| | [54] | Kano/TRIZ/QFD | √ | √ | √ | √ | | The paper identified user requirements and functional technical requirements with Kano and QFD. TRIZ was employed to propose design solutions for a four-wheel vehicle. |
| | [55] | Kano/QFD/PCM | √ | √ | √ | | √ | The paper employed Kano and QFD to identify user requirements and design factors and utilized PCM to evaluate design solutions. |
| | [56] | QFD/AHP/ARIZ | √ | √ | √ | √ | | The paper applied AHP and QFD to analyze the user requirements and technological features of disinfection and epidemic prevention robots. ARIZ proposed the design scheme. |
| | [57] | EGM/GRA/QFD | √ | √ | √ | | | The paper acquired user requirements for wickerwork lamps by EGM and GRA. QFD transformed the requirements into technical characteristics, leading to new design solutions. |
| Three or more Varieties | [58] | Kano/HOQ/QFD/TRIZ/ISM/DSM | √ | √ | √ | √ | | The innovative design of temporary refugee housing was achieved by employing the combined methods in this study. |
| | [59] | KJ/Kano/QFD/TRIZ/FEA | √ | √ | √ | √ | √ | The paper identified user and functional technical requirements by KJ, Kano, and QFD. TRIZ proposed the solutions, evaluated by FEA. |
| | [60] | FMEA/QFD/TRIZ/LCA/TOPSIS | √ | √ | √ | √ | √ | The paper employed FMEA and QFD to analyze user and technical requirements. Design solutions were proposed by TRIZ and evaluated through LCA and TOPSIS. |
| | [61] | SD/PCA/TRIZ/AHP | √ | | | √ | √ | The paper employed SD and PCA to analyze user requirements, proposed design solutions using TRIZ and evaluated the solutions by AHP. |
| | [62] | Kano/AHP/DEMATEL/QFD | √ | √ | √ | | | The paper employed Kano and AHP to investigate user requirements. DEMATE and QFD established the relationship between user requirements and design factors. |
| | [63] | Kano/AHP/QFD/PCM | √ | √ | √ | | √ | The paper identified user requirements with Kano and AHP, transformed the requirements into functional requirements using QFD, and evaluated the proposed solution by PCM. |
| | [64] | EGM/QTT1/TRIZ/AHP | √ | | | √ | √ | The paper employed EGM and QTT1 to analyze user requirements, proposed design solutions using TRIZ and evaluated the solutions by AHP. |
| | [65] | KJ/Kano/FAST/QFD | √ | √ | √ | | | The paper identified user requirements with KJ and Kano, decomposed the product functionalities using FAST, and applied QFD to establish the correlation between user needs and product features. |

they have integrated multiple theoretical methods. In reference [59], the KJ method and Kano survey were employed to analyze user requirements for Smart Neck Helmets (S1). And QFD was used to prioritize the importance of functional requirements and engineering parameters, providing a basis for analyzing design conflicts (S2, S3). Subsequently, TRIZ was applied to propose a new design solution (S4). Finally, the scheme was evaluated by FEA (S5). In

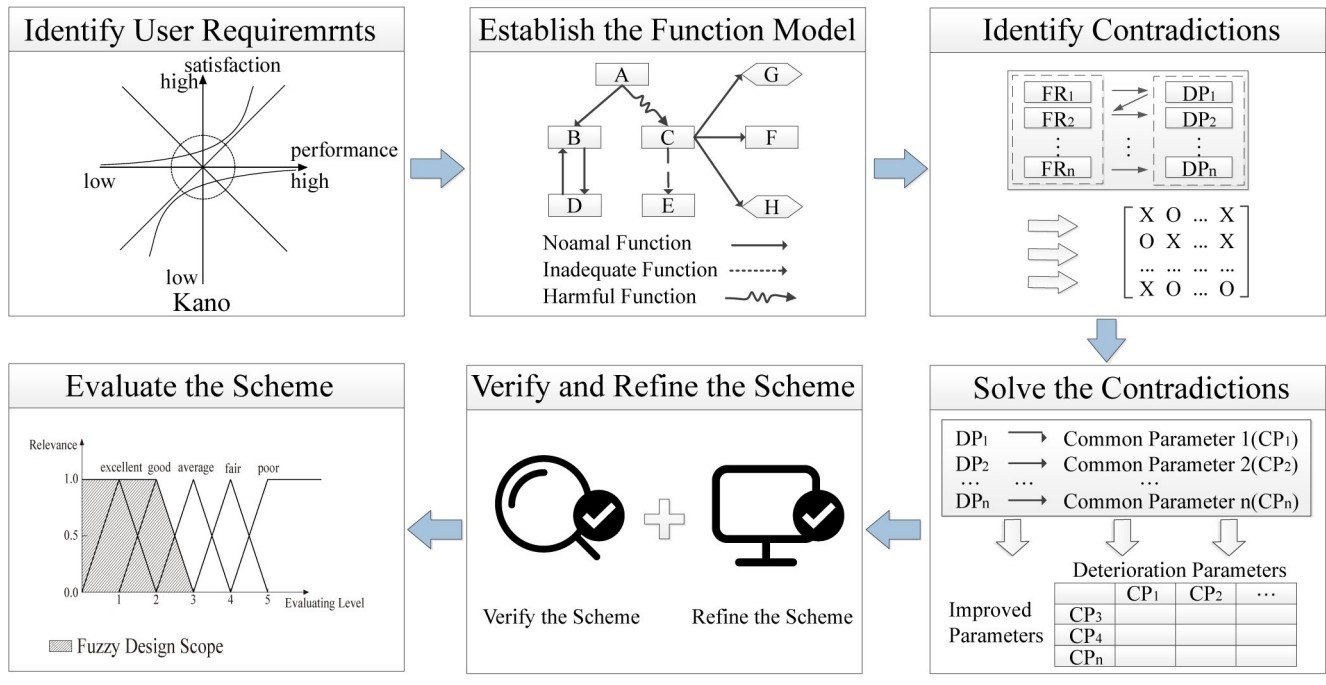

**Fig 3. The innovation design process based on Kano, AD, and TRIZ.**

reference [60], FMEA was utilized to capture user requirements for pressure relief valves. Then, QFD was employed to convert user requirements into functional and technical requirements, and TRIZ was used to develop a new valve structure solution. The solution was evaluated using LCA and Fuzzy TOPSIS. Although the studies guided the entire process of product innovation design, they necessitated the utilization of five different research theories and methods, which had potential drawbacks. First, it required readers to have a broad theoretical foundation to understand the design process and comprehend the research articles. Moreover, the multitude of methods increased the complexity of the design practice, which limited the practical application of these methods.

The paper proposed a new product innovation design process. This new process has two advantages. Firstly, this method can guide the entire process of product innovation design. Additionally, the new process incorporates only three theoretical methods, greatly reducing the learning burden for readers, while also lowering the difficulty and complexity of design practice.

## Method

The theoretical approach proposed in this paper consists of six main parts: clarifying user requirements, establishing a product function model, analyzing the conflicts between functional requirements and design parameters, resolving the conflicts based on TRIZ, validating and refining the design solution, and evaluating the scheme. The process is illustrated in Fig 3.

### Clarify user requirements

The Kano model is applied to identify user needs and classify them into different types, following the steps of designing the questionnaire, collecting responses, and processing data. This

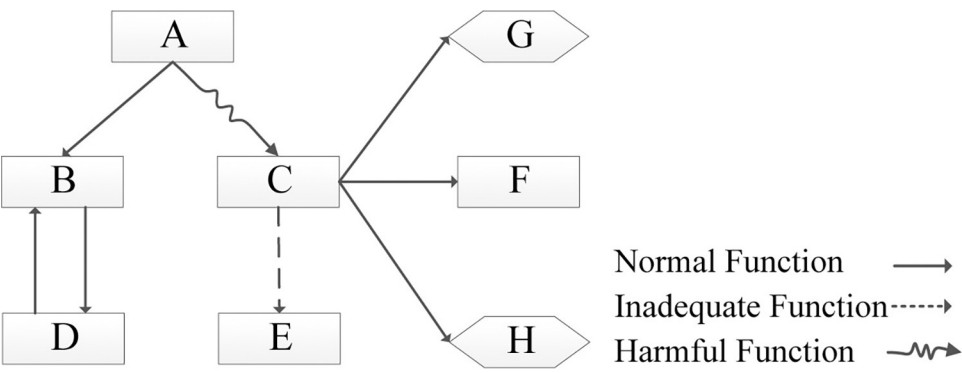

**Fig 4. Establish the function model.**

step provides a basis for mapping user needs to functional requirements and enhances accuracy.

## Establish the function model

After clarifying user needs through the Kano model, in order to improve the accuracy of mapping functional requirements to design parameters, it is necessary to have a systematic understanding of the structural principles of the product. The paper will establish a functional model to decompose the functional structure of the product [66]. The functional model describes the relationships and interactions between components or parts of the product in a graphical manner [67], as shown in Fig 4.

## Analyze contradictions between product functions and design parameters

This process consists of three main steps.

Firstly, the functional requirements are decomposed based on the user needs identified by Kano.

Secondly, the mapping relationship between functional requirements and design parameters is established based on the functional model, as shown in Table 3. The design equation can be represented as Eq (1), where X indicates the influence of the structure on the function, and O indicates no influence of the structure on the function.

$$\begin{Bmatrix} FR_1 \\ FR_2 \end{Bmatrix} = \begin{bmatrix} X & O \\ O & X \end{bmatrix} \cdot \begin{Bmatrix} DP_1 \\ DP_2 \end{Bmatrix} \tag{1}$$

Thirdly, identify contradictions between product functional requirements and design parameters. According to the independence axiom in AD theory, the coupling of the design matrix is determined. Take Eq (2) as an example. The form of the matrix indicates that it is a coupling matrix. Three groups of conflicts exist between functional requirements and design

**Table 3. Build the mapping relationship between functional requirements and design parameters.**

| Functional Requirements (FRs) | Design Parameters (DPs) |
|---|---|
| $FR_1$ | $DP_1$ |
| $FR_2$ | $DP_2$ |

parameters, namely $FR_1$ and $DP_3$, $FR_3$ and $DP_1$, and $FR_3$ and $DP_2$. To decouple the matrix, these three conflicts are required to be solved.

$$\begin{Bmatrix} FR_1 \\ FR_2 \\ FR_3 \end{Bmatrix} = \begin{bmatrix} X\ O\ X \\ O\ X\ O \\ X\ X\ X \end{bmatrix} \cdot \begin{Bmatrix} DP_1 \\ DP_2 \\ DP_3 \end{Bmatrix} \qquad (2)$$

## Resolve conflicts based on TRIZ

The process mainly consists of three steps.

Firstly, complete the conversion of general parameters. In this step, the product components involved in the contradiction between function and structure need to be transformed into engineering parameters according to the definition in TRIZ theory, and the improvement factors and deterioration factors should be identified.

Secondly, select appropriate inventive principles and resolve contradictions. Based on the results of factor transformation in Step 1, refer to the Achshuler conflict matrix to find the inventive principle corresponding to the intersection of the improved and deteriorated parameters. Compare the obtained inventive principles with the design objectives and select the appropriate ones to resolve the design conflicts.

## Validate and refine the design scheme

Verify the design scheme using the independence axiom in AD theory to determine its rationality. If the design scheme is deemed reasonable, further improvements can be made to enhance its effectiveness.

## Evaluate the design scheme

After confirming that the design scheme complies with the independent axioms of AD and refining the design scheme, evaluate the design scheme by the information axioms in AD, comparing the amount of information in the scheme with the original product. Firstly, convert the user requirements obtained through Kano analysis into evaluation system indicators. Secondly, divide the evaluation levels of each indicator and determine the design scope based on user requirements. Thirdly, evaluate the scheme through expert evaluation. Finally, take the triangular fuzzy function as the membership function to convert linguistic terms into corresponding fuzzy numbers [68], draw the function graph, and calculate the amount of information according to Eq (3).

$$I = log_2 \left( \frac{fuzzy\ design\ scope}{fuzzy\ public\ scope} \right) \qquad (3)$$

## Case study

The self-balancing two-wheeled vehicle, as a means of transportation, has gained popularity among people in modern society. It has become common for college students to see them taking self-balancing two-wheeled vehicles to travel around the campus. This paper takes the design of a self-balancing two-wheeled vehicle as a practical case study to demonstrate the feasibility and effectiveness of the new process.

**Table 4. Analysis of user requirements in Kano.**

| The first-level user requirements | The second-level user requirements | Kano attributes | Better coefficient | Worse coefficient |
|---|---|---|---|---|
| Commuting functionality | Ease of standing on the vehicle | M | 11.90% | -88.10% |
| | Riding speed | I | 35.71% | -26.19% |
| Safety | Stable structures | M | 11.9% | -95.24% |
| | Responsive braking system | M | 16.67% | -80.95% |
| Comfort | Good shock absorption | A | 83.33% | -40.48% |
| Portability | Lightweight | A | 80.95% | -19.05% |
| | Handle for carrying | A | 61.9% | -11.9% |
| | Reduced size when idle | A | 83.33% | -7.14% |
| Maintainability | Simple structures for easy maintenance | O | 45.24% | -57.14% |
| Aesthetics | Aesthetically pleasing appearance | O | 64.29% | -59.52% |

### Identify user requirements for the self-balancing two-wheeled vehicle

Through expert evaluation, the user requirements for the self-balancing two-wheeled vehicle are determined by Kano analysis regarding commuting functionality, safety, comfort, portability, maintainability, and aesthetics. Based on the final calculation results, Must-Requirements include "ease of standing on the vehicle", "stable structures", and "responsive braking system". One-Dimensional Requirements include "simplicity of structure for easy maintenance" and "aesthetically pleasing appearance". Attractive Requirements include "good shock absorption", "lightweight", "handle for carrying", and "reduced size when idle". Indifferent Requirement is "riding speed". All the calculation results are presented in Table 4.

### Establish the functional model of the self-balancing two-wheeled vehicle

Through function analysis, the structural principles of the two-wheeled balancing vehicle, as well as the relationships and interactions among its components, can be determined, as shown in Fig 5. This lays the foundation for establishing the mapping relationship between FRs and DPs.

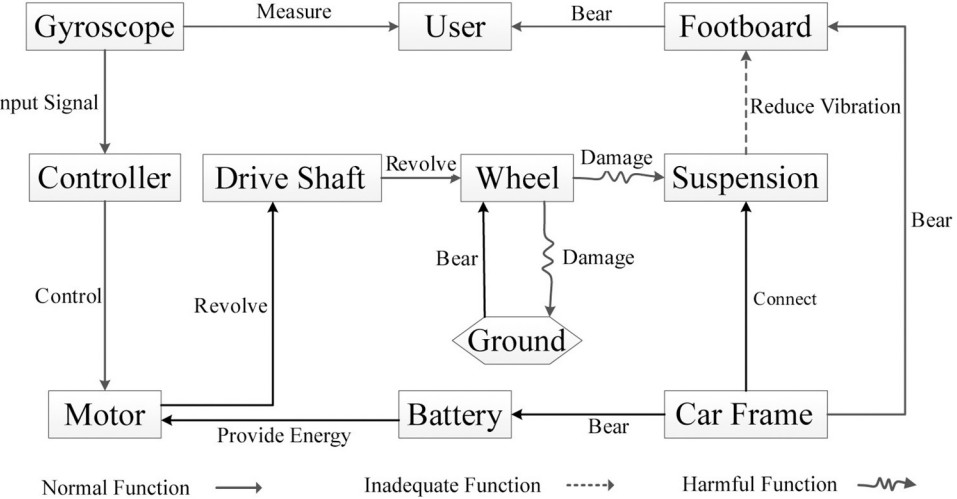

**Fig 5. Establish the function model of a self-balancing two-wheeled vehicle.**

## Analyze the conflicts between the functional requirements and design parameters

Firstly, decompose the functional requirements based on the user requirements identified by Kano, as shown in Fig 6.

Secondly, establish the mapping relationship between functional requirements and design parameters based on the functional model. Refer to Table 5 for details.

Thirdly, analyze the conflicts between functional requirements and design parameters based on the independence axiom. According to the mapping results in Table 5, the equations between portable self-balancing scooter functional requirements and product structure are listed, resulting in the design equation as shown in Eq (4). From the design matrix, it can be observed that there are 5 groups of conflicts.

$$
\begin{Bmatrix} FR_{11} \\ FR_{12} \\ FR_{21} \\ FR_{22} \\ FR_{31} \\ FR_{41} \\ FR_{42} \\ FR_{43} \\ FR_{51} \\ FR_{61} \end{Bmatrix} =
\begin{bmatrix}
X & O & O & O & O & O & O & O & O & O \\
O & X & O & O & O & O & O & O & O & O \\
O & O & X & O & O & X & O & X & X & O \\
O & O & O & X & O & O & O & O & O & O \\
O & O & O & O & X & O & O & O & O & O \\
O & O & O & O & O & X & O & O & O & O \\
O & O & O & O & O & O & X & O & O & X \\
X & O & O & O & O & O & O & X & O & O \\
O & O & X & O & O & O & O & O & X & X & O \\
O & O & O & O & O & O & O & O & O & X
\end{bmatrix} \cdot
\begin{Bmatrix} DP_{11} \\ DP_{12} \\ DP_{21} \\ DP_{22} \\ DP_{31} \\ DP_{41} \\ DP_{42} \\ DP_{43} \\ DP_{51} \\ DP_{61} \end{Bmatrix}
\tag{4}
$$

There exists a conflict between "stable structures($FR_{21}$)" and "material($DP_{41}$)". Achieving structural stability requires high material strength, but high-strength materials may have a higher density, which would increase the weight of the self-balancing two-wheeled vehicle.

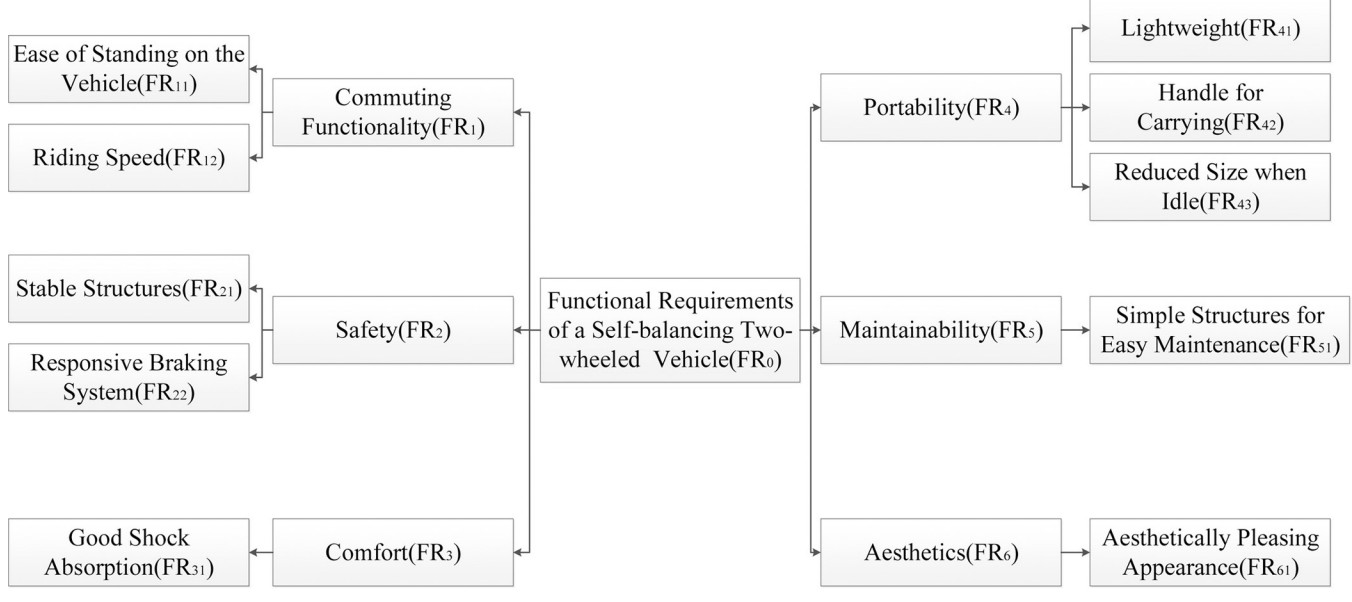

**Fig 6. The decomposition of the functional requirements.**

**Table 5. The mapping relationship between functional requirements and design parameters.**

| Functional Requirements (FRs) | Design Parameters (DPs) |
|---|---|
| Ease of standing on the vehicle($FR_{11}$) | Footboard area($DP_{11}$) |
| Riding speed($FR_{12}$) | Transmission structure and wheels($DP_{12}$) |
| Stable structures($FR_{21}$) | Car frame ($DP_{21}$) |
| Responsive braking system($FR_{22}$) | Gyroscope and controller($DP_{22}$) |
| Good shock absorption ($FR_{31}$) | Suspension structure($DP_{31}$) |
| Lightweight($FR_{41}$) | Material ($DP_{41}$) |
| Handle for carrying($FR_{42}$) | Lifting structure($DP_{42}$) |
| Reduced size when idle ($FR_{43}$) | Folding structures($DP_{43}$) |
| Simple structures for easy maintenance ($FR_{51}$) | Car frame($FR_{51}$) |
| Aesthetically pleasing appearance ($FR_{61}$) | Appearance ($FR_{61}$) |
| Ease of standing on the vehicle($FR_{11}$) | Footboard area($DP_{11}$) |

There exists a conflict between " stable structures($FR_{21}$)" and "folding structures($DP_{43}$)". While the inclusion of a folding structure may reduce the size when idle, it may also introduce movable components that compromise the structural stability.

There exists a conflict between "handle for carrying($FR_{42}$)" and "appearance($DP_{61}$)". Adding a handle would make it more convenient for users to carry the self-balancing two-wheeled vehicle, but it may affect the product's overall aesthetics.

There exists a conflict between " reduced size when idle ($FR_{43}$)" and "footboard area ($DP_{11}$)". Reducing the product size may result in a smaller footboard area, which could impact the riding experience for users.

There exists a conflict between "simple structures for easy maintenance($FR_{51}$)" and "folding structures($DP_{43}$)". The inclusion of a folding structure would increase the complexity of the vehicle's structure, making maintenance more challenging.

## Resolve design issues of the self-balancing two-wheeled vehicle based on TRIZ

Firstly, convert the factors of the self-balancing two-wheeled vehicle into general engineering parameters. Extract the factors that need improvement and the factors that may worsen from the five groups of conflicts, and transform them into the general parameters, as shown in Table 6.

Secondly, select appropriate invention principles. Through the general parameters obtained from Table 6, the paper refers to the Achshuler conflict matrix to find the corresponding invention, as shown in Table 7. More than twenty invention principles are obtained based on the conflict matrix. After careful selection, the paper chooses principles of invention numbered 2, 5, 15, and 7 as the guiding methods for designing the self-balancing two-wheeled vehicle.

**Table 6. Factors transformation.**

| Serial number | Factors that need to be improved | | Factors that will worsen | |
|---|---|---|---|---|
| | Factors | General parameters | Factors | General parameters |
| 1 | Stable structures | Stability of the structure(NO.13) | Weight | The weight of a stationary object(NO.2) |
| 2 | Reduced size when idle | Adaptability and versatility(NO.35) | Stable structures | Stability of the structure(NO.13) |
| 3 | Handle for carrying easily | Operability(NO.33) | Appearance | Shape(NO.12) |
| 4 | Reduced size when idle | The volume of a stationary object(NO.8) | Footboard area | The area of a moving object(NO.5) |
| 5 | Added folding structure | The number of materials or objects(NO.26) | Easy maintenance | Maintainability(NO.34) |

**Table 7. The invention principles found in the Achshuler conflict matrix.**

| Improved parameters | Deterioration parameter | | | | |
|---|---|---|---|---|---|
| | The weight of a stationary object(NO.2) | Stability of the structure (NO.13) | Shape (NO.12) | The area of a moving object(NO.5) | Maintainability (NO.34) |
| Stability of the structure(NO.13) | 26,29,1,40 | / | 22,1,18,4 | 2,11,13 | 2,35,10,16 |
| Adaptability and versatility (NO.35) | 9,15,29,16 | 35,30,14 | 15,37,1,8 | 35,30,29,7 | 1,16,7,4 |
| Operability(NO.33) | 6,13,1,25 | 32,35,30 | 15,34,29,28 | 1,17,13,16 | 12,26,1,32 |
| The volume of a stationary object (NO.8) | 35,10,19,14 | 34,28,35,40 | 7,2,35 | / | 1 |
| The number of materials or objects(NO.26) | 27,26,18,35 | 15,2,17,40 | 35,14 | 15,14,29 | 2,32,10,25 |

Thirdly, utilize the invention principles to resolve design issues.

Above all, according to the invention principle No.2, the principle of extraction means that removing the part or attribute that has a negative effect is extracted from the object. The power transmission shaft component in the power system is extracted, thereby reducing the complexity and difficulty of the folding wheel structure.

Next, the invention principle No.5 is used for combination, which involves combining or merging the same objects or related operations in space or time. Combine the motor that generates power with the moving wheel, namely utilize hub motors to provide motive force for a self-balancing two-wheeled vehicle. The permanent magnet hub motor designed and researched by Ai Dong et al. can be a good choice, characterized by good stability, lightweight, and compact design [69].

Furthermore, the invention principle No.15, the dynamic principle, is meant to divide an object and make each of its parts alter its relative positions. To achieve folding function, certain components of the self-balancing scooter are allowed to change their relative positions. During this process, it is necessary to set up two rotating mechanisms to achieve the folding of the footboards and the wheels. A hinge is used between the handle and the footboard to achieve rotational folding of the footboard. The shape of the handle is designed to limit the footboard's rotation angle, ensuring the product structure's stability while riding. Moreover, it ensures the footboards can be quickly and easily folded when users need to carry the vehicle. A connecting shaft is employed between the wheel with a hub motor and the footboard. To realize the folding of wheels, a fixed joint is utilized to connect the connecting shaft with the wheel, while a rotating joint is adopted to connect the connecting shaft with the footboard. The structure enables the wheel to rotate onto the two-dimensional plane where the footboard is located, further reducing the overall volume of the product after folding.

Eventually, the invention principle No.7, the nesting principle, is employed to allow one object to pass through or be within the cavity of another object. To ensure the stability of the vehicle structure during riding and the simplicity of folding operation, a self-locking system needs to be designed. Two connecting shafts with different shape features are nested between the connecting shaft and the footboard to achieve dual functionality. During the operation of the vehicle, a square axis is used to connect the connecting shaft and the footboard to prevent relative displacement between the footboard and the wheel. When folding the self-balancing vehicle, the square axis is replaced with a round axis, so that the connecting shaft can act as a pivot. These two connecting shafts can be interchanged through buttons, card slots, and springs.

By comprehensively applying the inventive principles, a folding structure that meets the requirements is obtained. The schematic diagram of this structure is shown in Fig 7. As the two-wheeled self-balancing vehicle is symmetrical, only the right side is shown in the figure.

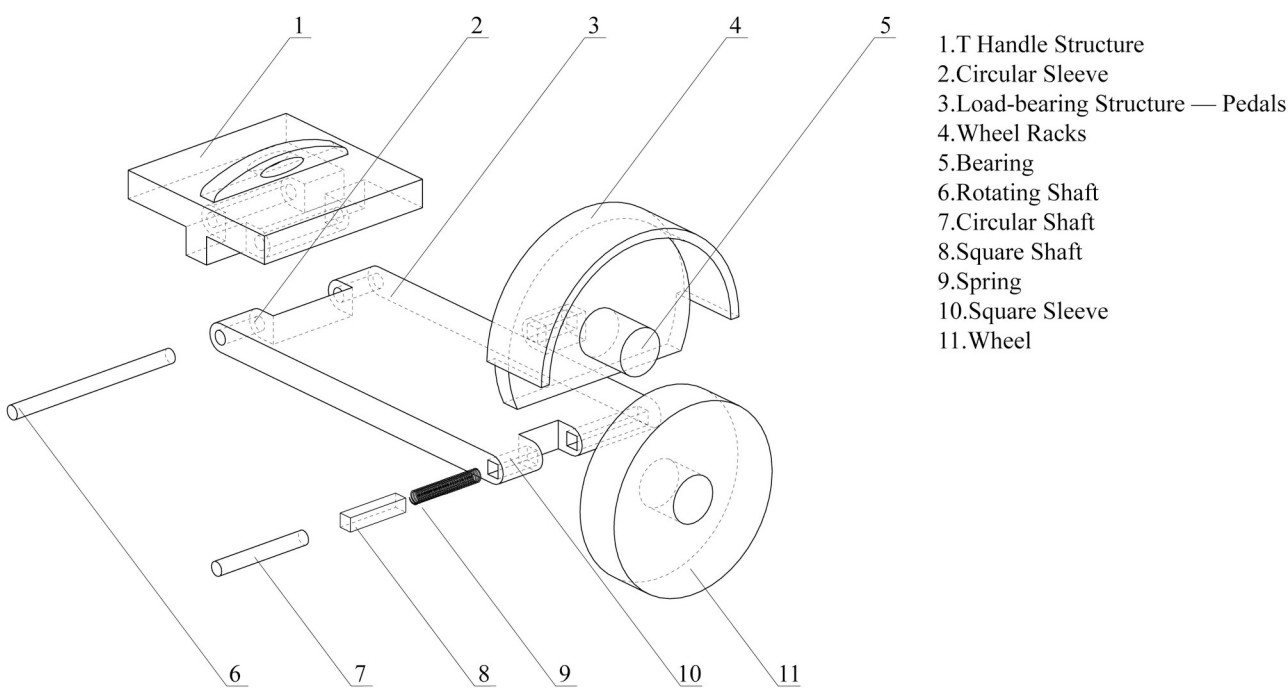

**Fig 7. The structural explosion diagram of the product in the design scheme.**

## Validate and refine the design scheme

Firstly, the rationality of the design proposal is validated using the independence axiom of the AD theory. The mapping relationship between the functional requirements and design parameters of the self-balancing two-wheeled vehicle in the design scheme is established, as shown in Table 8. The design matrix, as shown in Eq (5), indicates a non-coupled design according to the definition of the AD theory, suggesting that the design scheme is theoretically reasonable.

$$
\begin{Bmatrix} FR_{11} \\ FR_{12} \\ FR_{21} \\ FR_{22} \\ FR_{31} \\ FR_{41} \\ FR_{42} \\ FR_{43} \\ FR_{51} \\ FR_{61} \end{Bmatrix} = \begin{bmatrix} X\,O\,O\,O\,O\,O\,O\,O\,O\,O \\ O\,X\,O\,O\,O\,O\,O\,O\,O\,O \\ O\,O\,X\,O\,O\,O\,O\,O\,O\,O \\ O\,O\,O\,X\,O\,O\,O\,O\,O\,O \\ O\,O\,O\,O\,X\,O\,O\,O\,O\,O \\ O\,O\,O\,O\,O\,X\,O\,O\,O\,O \\ O\,O\,O\,O\,O\,O\,X\,O\,O\,O \\ O\,O\,O\,O\,O\,O\,O\,X\,O\,O \\ O\,O\,O\,O\,O\,O\,O\,O\,X\,O \\ O\,O\,O\,O\,O\,O\,O\,O\,O\,X \end{bmatrix} \cdot \begin{Bmatrix} DP'_{11} \\ DP'_{12} \\ DP'_{21} \\ DP'_{22} \\ DP'_{31} \\ DP'_{41} \\ DP'_{42} \\ DP'_{43} \\ DP'_{51} \\ DP'_{61} \end{Bmatrix} \qquad (5)
$$

Secondly, refine the design scheme based on the mechanical structure in the new design scheme. The design schematics of the self-balancing two-wheeled vehicle are shown in Fig 8. This self-balancing two-wheeled is different from traditional ones. It can serve as a means of transportation when the user needs to travel from one place to another, and can also be folded

**Table 8. The new mapping relationship between functional requirements and design parameters.**

| Functional Requirements (FRs) | Design Parameters ($DP'_s$) |
|---|---|
| Ease of standing on the vehicle($FR_{11}$) | Footboard area ($DP'_{11}$) |
| Riding speed($FR_{12}$) | Hub motors and wheels ($DP'_{12}$) |
| Stable structures($FR_{21}$) | Self-locking structure ($DP'_{21}$) |
| Responsive braking system($FR_{22}$) | Gyroscope and controller ($DP'_{22}$) |
| Good shock absorption ($FR_{31}$) | Suspension structure ($DP'_{31}$) |
| Lightweight($FR_{41}$) | Materials with high strength and low density ($DP'_{41}$) |
| Handle for carrying($FR_{42}$) | T handle structure ($DP'_{42}$) |
| Reduced size when idle ($FR_{43}$) | Rotating shaft ($DP'_{43}$) |
| Simple structures for easy maintenance ($FR_{51}$) | Car frame ($DP'_{51}$) |
| Aesthetically pleasing appearance ($FR_{61}$) | Appearance ($DP'_{61}$) |

by the user when not in use. The folding steps are shown in Fig 9. Taking into account both commuting and portability in terms of ergonomics, the product size is determined as depicted in Fig 10. The product has a simple and compact appearance, with colors mainly black and gray. And blue vehicle lights are used to enhance its technological sense. The main material options for the product body include aluminum alloy, magnesium alloy, or carbon fiber, which ensure strength while maintaining lightweight quality. These materials possess high plasticity and mature manufacturing processes, including sheet metal processing, spinning, stamping, deep drawing, and superplastic forming. In addition, the surface can be changed with spray painting or powder coating to increase corrosion resistance and scratch resistance.

## Evaluation of the self-balancing two-wheeled vehicle design scheme

First, the user requirements obtained through Kano analysis are transformed into evaluation system indicators, namely, commuting functionality(A), safety(B), comfort(C), portability(D), maintenance complexity(E), and aesthetics(F).

Next, divide the evaluation levels of each indicator and determine the design scope on user requirements. A-D and F are divided into five evaluation levels: poor, fair, average, good, and excellent, while E is divided into three evaluation levels: simple, average, and complex. The design ranges of each indicator are determined based on user requirements. The design range for A-D and F is good and above, while the design range for E is average and below.

(a)                                          (b)

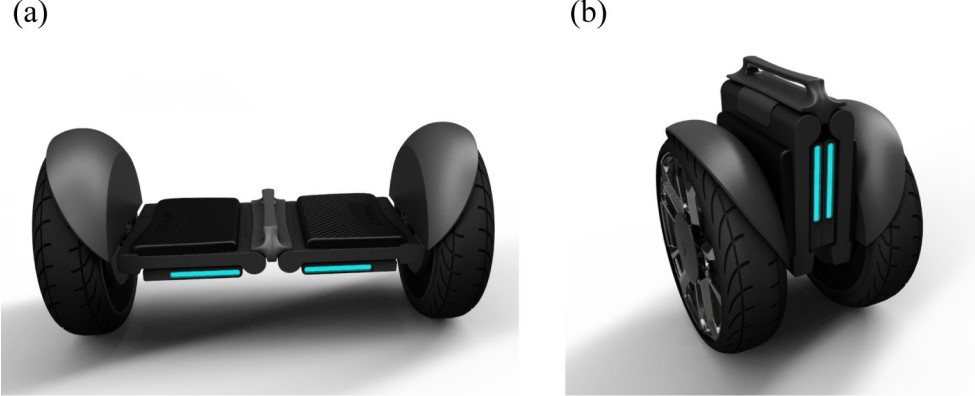

**Fig 8. The design schematics of the portable self-balancing two-wheeled vehicle.** (a) This figure shows the deployment state of a portable two-wheeled balancing vehicle. (b). This figure shows the folding state of a portable two-wheeled balancing vehicle.

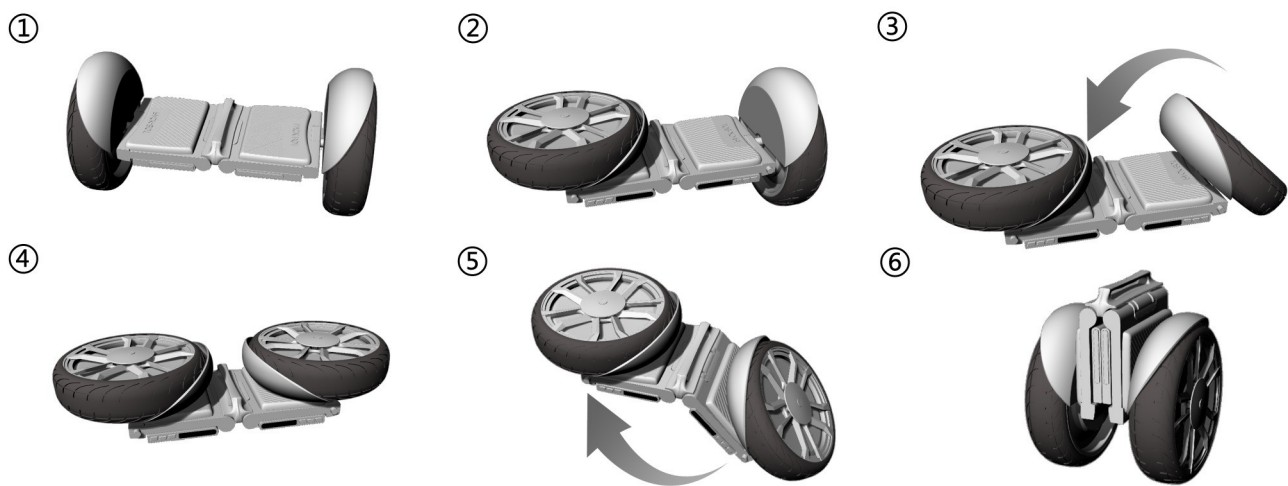

**Fig 9. The operation procedures of folding the self-balancing two-wheeled vehicle.**

Then, through expert evaluation, the two self-balancing two-wheeled vehicles are assessed, and the evaluation results are shown in Table 9.

Finally, calculate the information amount based on the plotted function graph. Take the commuting function (A) as an example; the plotted function graph is shown in Fig 11. Eq (6) and Eq (7) show the information calculation processes. All the evaluation results are in Table 10.

$$I_{(the\ design\ scheme)} = log_2\left(\frac{fuzzy\ design\ scope}{fuzzy\ public\ scope}\right) = log_2\left[\frac{(3+2)\times 1\times\frac{1}{2}}{(3-1)\times 1\times\frac{1}{2}}\right] = 1.3219 \tag{6}$$

$$I_{(the\ original\ vehicle)} = log_2\left(\frac{fuzzy\ design\ scope}{fuzzy\ public\ scope}\right) = log_2\left[\frac{(3+2)\times 1\times\frac{1}{2}}{(3+1)\times 1\times\frac{1}{2}}\right] = 0.3219 \tag{7}$$

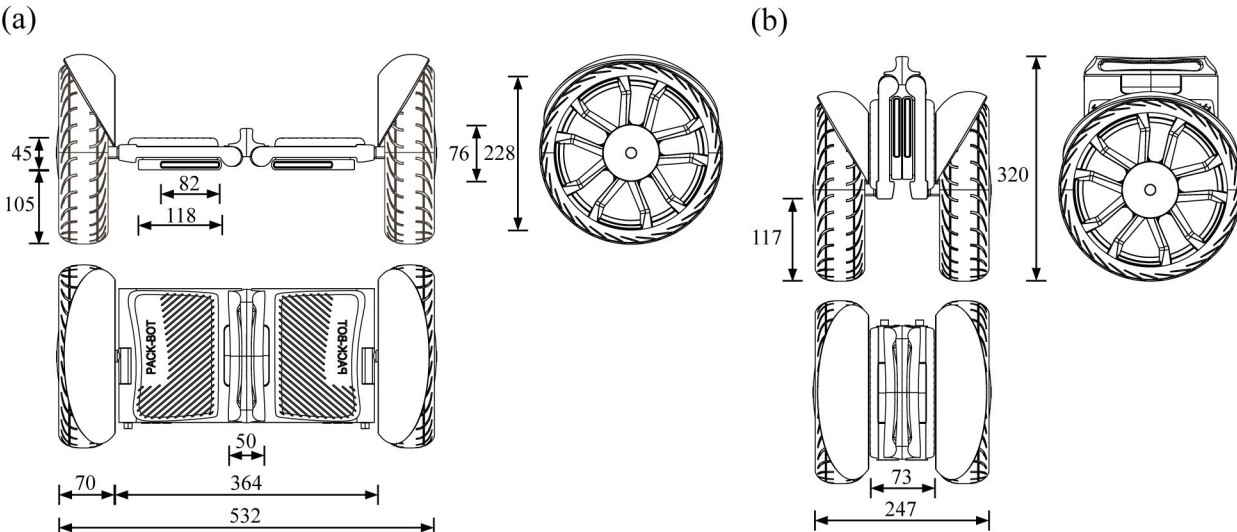

**Fig 10. Product size chart.** (a) This figure shows the product size of the self-balancing two-wheeled vehicle as it is deployed. (b) This figure shows the product size of the self-balancing two-wheeled vehicle as it is folded.

**Table 9. The evaluation results of two self-balancing two-wheeled vehicles.**

|  | Commuting functionality (A) | Safety (B) | Comfort (C) | Portability (D) | Maintenance complexity (E) | Aesthetics (F) |
|---|---|---|---|---|---|---|
| The design scheme | Good | Good | Good | Excellent | Average | Good |
| The original vehicle | Excellent | Good | Good | Average | Simple | Average |

## Discussion

### Contribution, significance, and limitations

This paper proposed a process for product innovation design by combining Kano, AD, and TRIZ. The new process was applied to the design practice of a self-balancing two-wheeled vehicle, which resulted in an improved design scheme. The final design output demonstrated that the proposed method could effectively guide the entire product innovation design process, helping designers understand user needs, analyze functional requirements, identify and resolve structural contradictions, and evaluate design solutions. In the new method, Kano's identification of user needs clarifies the design direction in the early stages and forms an evaluation model for design solutions in conjunction with AD during the design evaluation stage. The AD theory plays a role in transforming user needs, analyzing design contradictions, and evaluating design solutions in the new process. These theoretical methods fully demonstrate their effectiveness and characteristics in the new process, leading to a reduced complexity of integrated methods in theory and practice compared to previous studies [59, 60].

However, shortcomings in the new process can also be clearly identified during design practice. Firstly, the method proposed in this paper cannot effectively identify dynamically changing user requirements. Reference [70] combined Kano, grey relational analysis, and benchmarking theory to help identify and calculate dynamically changing user requirements and satisfaction. Additionally, using TRIZ to find innovative methods can lengthen the design cycle. Although TRIZ has summarized many parameters and inventive principles, it still relies on the designer's experience for practical application selection. In this process, the design team needs to go through several trial-and-error attempts to find the correct direction, which undoubtedly prolongs the design cycle. Reference [71] proposed utilizing artificial neural

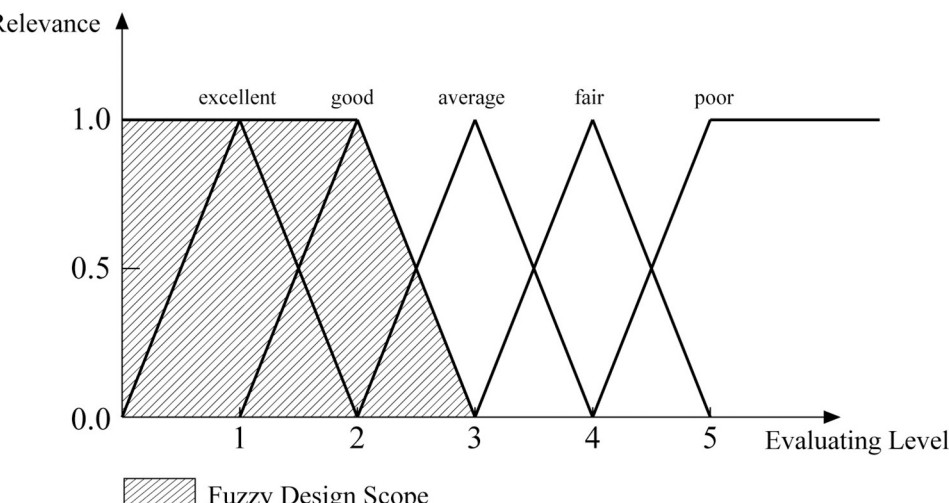

**Fig 11. The membership function curve of the commuting function(A).**

**Table 10. Calculation results of information quantity for each evaluation indicator.**

| Evaluation indicators | The design scheme | The original vehicle |
|---|---|---|
| Commuting functionality(A) | 1.3219 | 0.3219 |
| Safety(B) | 1.3219 | 1.3219 |
| Comfort(C) | 1.3219 | 1.3219 |
| Portability(D) | 0.3219 | 3.3219 |
| Maintenance complexity(E) | 1.3219 | 0.3219 |
| Aesthetics (F) | 1.3219 | 3.3219 |
| Total | 6.9314 | 9.9314 |

networks (ANN) to address this issue. In conclusion, there are still shortcomings in the new process proposed in this study. Further research should try to address the shortcomings through optimization methods or by considering other perspectives.

## Insights from managerial perspectives

Based on the above discussion, it can be concluded that the proposed method has limitations in both capturing dynamic user requirements and efficiently utilizing TRIZ. Appropriate management approaches can partially compensate for the drawbacks of new processes without increasing the complexity of theoretical methods in practical implementation. The following will explain the design management approach of agile development and discuss the potential of using this management approach to address the drawbacks based on literature research and successful design cases.

Agile development originally referred to a design and management approach for software development that emphasizes iteration and incremental progress [72]. It underlines rapid feedback, flexibility, and collaboration, focusing on team communication and cooperation to adapt to changing requirements and rapidly iterate product updates [73]. This design management approach has gradually been extended to physical product design, known as agile product development [74–76].

The agile product development management approach requires constant feedback from users during the product design process to adapt to user needs and adjust design plans promptly. Simultaneously, technical development for implementing product functionalities should be conducted while conceptualizing design plans, with regular feedback provided [77]. The application of the agile product development management approach in industrial case studies, as demonstrated in the literature [77–79], has proven the effectiveness of this management approach in responding to dynamic user needs and rapidly iterating product technology plans. Many commercially successful products have been developed using this design management approach, such as Tesla's Model 3. Tesla's design team closely collaborated with potential users during the design process, rapidly collecting feedback on requirements and opinions on automotive interior design and promptly adjusting design plans. Moreover, due to the parallel design and development management approach, Tesla could proceed with the release and sales of Model 3 without any delays, even in the event of design plan adjustments.

In summary, the agile product development management approach facilitates tracking changes in user requirements and rapid iteration of product solutions. This effectively compensates for the shortcomings of the proposed method process in this paper. It provides a research direction for the subsequent optimization of the new approach.

## Conclusion

The proposed design method combines Kano, AD, and TRIZ, covering the processes and steps of researching user needs, transforming needs into functional requirements, analyzing design problems, solving design problems, and evaluating design solutions. It guided the entire design process with fewer theoretical approaches, reducing the complexity of theory and practice. The design practice of a self-balancing two-wheeled vehicle demonstrated the new method's effectiveness. However, there are still limitations and shortcomings in this study. Firstly, the proposed theoretical method weakens the ability to capture dynamically changing user needs. Additionally, selecting appropriate engineering parameters and inventive principles from TRIZ relies on personal experience and multiple trial and error. From a management perspective, when designing practice according to the new process, combining the agile product development management approach may compensate for the new process's shortcomings without increasing theoretical complexity. Future research will attempt to optimize the proposed method process from the perspectives of management and other disciplines, enhancing the ability to follow up on changes in user needs and improving the accuracy of selecting engineering parameters and inventive principles.

## Supporting information

**S1 File. The minimal underlying data set.**
(PDF)

## Acknowledgments

Thanks to Professor Liu for her academic guidance. Thanks to Jin Li and Ziqian Zhou for their research.

## Author Contributions

**Conceptualization:** Hui Rong, Wei Liu.

**Data curation:** Hui Rong, Jin Li.

**Investigation:** Jin Li, Ziqian Zhou.

**Methodology:** Hui Rong, Wei Liu, Jin Li, Ziqian Zhou.

**Visualization:** Hui Rong, Jin Li.

**Writing – original draft:** Hui Rong.

**Writing – review & editing:** Hui Rong, Wei Liu.

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
