## [Decision Letter · Decision Letter 0]

5 Sep 2023

PONE-D-23-21362Product innovation design process combined Kano and TRIZ with AD: case studyPLOS ONE

Dear Dr. liu,

Thank you for submitting your manuscript to PLOS ONE. After careful consideration, we feel that it has merit but does not fully meet PLOS ONE’s publication criteria as it currently stands. Therefore, we invite you to submit a revised version of the manuscript that addresses the points raised during the review process.

Please submit your revised manuscript by Oct 20 2023 11:59PM. If you will need more time than this to complete your revisions, please reply to this message or contact the journal office at plosone@plos.org. Please include the following items when submitting your revised manuscript:A rebuttal letter that responds to each point raised by the academic editor and reviewer(s). You should upload this letter as a separate file labeled 'Response to Reviewers'.A marked-up copy of your manuscript that highlights changes made to the original version. You should upload this as a separate file labeled 'Revised Manuscript with Track Changes'.An unmarked version of your revised paper without tracked changes. You should upload this as a separate file labeled 'Manuscript'.

We look forward to receiving your revised manuscript.

Kind regards,

Mazyar Ghadiri Nejad, Ph.D.

Academic Editor

PLOS ONE

“Professor Wei Liu from Nanjing Forestry University is the sponsor of the paper research submitted this time. She played the roles of project management and writing guidance in this study.”

4. "PLOS requires an ORCID iD for the corresponding author in Editorial Manager on papers submitted after December 6th, 2016. Please ensure that you have an ORCID iD and that it is validated in Editorial Manager. To do this, go to ‘Update my Information’ (in the upper left-hand corner of the main menu), and click on the Fetch/Validate link next to the ORCID field. This will take you to the ORCID site and allow you to create a new iD or authenticate a pre-existing iD in Editorial Manager. Please see the following video for instructions on linking an ORCID iD to your Editorial Manager account: https://www.youtube.com/watch?v=_xcclfuvtxQ

6. We note that Figures 3, 7, 8, 9 and 10 in your submission contain copyrighted images. All PLOS content is published under the Creative Commons Attribution License (CC BY 4.0), which means that the manuscript, images, and Supporting Information files will be freely available online, and any third party is permitted to access, download, copy, distribute, and use these materials in any way, even commercially, with proper attribution. For more information, see our copyright guidelines: http://journals.plos.org/plosone/s/licenses-and-copyright.

A. You may seek permission from the original copyright holder of Figures 3, 7, 8, 9 and 10 to publish the content specifically under the CC BY 4.0 license.

B. If you are unable to obtain permission from the original copyright holder to publish these figures under the CC BY 4.0 license or if the copyright holder’s requirements are incompatible with the CC BY 4.0 license, please either i) remove the figure or ii) supply a replacement figure that complies with the CC BY 4.0 license. Please check copyright information on all replacement figures and update the figure caption with source information. If applicable, please specify in the figure caption text when a figure is similar but not identical to the original image and is therefore for illustrative purposes only.

Reviewers' comments:

Reviewer's Responses to Questions

**Comments to the Author**

1. Is the manuscript technically sound, and do the data support the conclusions?

Reviewer #1: Yes

Reviewer #2: Yes

2. Has the statistical analysis been performed appropriately and rigorously? 

Reviewer #1: Yes

Reviewer #2: Yes

3. Have the authors made all data underlying the findings in their manuscript fully available?

Reviewer #1: Yes

Reviewer #2: Yes

4. Is the manuscript presented in an intelligible fashion and written in standard English?

Reviewer #1: Yes

Reviewer #2: Yes

5. Review Comments to the Author

Reviewer #1: 1. How do you ensure that the integration of Axiomatic Design, Kano, and TRIZ is seamless and that the strengths of each theory are effectively leveraged without creating conflicts or redundancies?

2. Could there be challenges in accurately classifying and prioritizing user needs using the Kano model? How do you address potential variations in user needs within different market segments?

3. While the method aims to enhance innovation capability, could the incorporation of TRIZ for solving problems potentially lead to extended design iteration cycles? How can you balance rapid iteration with the thoroughness of TRIZ analysis?

4. Given that this integrated approach requires familiarity with multiple theories, how will designers be trained to effectively use and integrate Axiomatic Design, Kano, and TRIZ? How do you manage the potential learning curve for design teams?

5. The method is proposed as applicable to various industries. How do you account for the differing nature of products, markets, and user needs across these diverse sectors?

6. How do you balance the potential benefits of the integrated approach with the associated resource costs?

7. While the method's effectiveness will be demonstrated through the self-balancing two-wheeled vehicle project, how can you ensure that the success is not overly specific to this case and can be generalized to other product types?

8. TRIZ's toolbox is dynamic and evolves. How do you ensure that your method stays up-to-date with the latest TRIZ tools and techniques for problem-solving?

9. As technological advancements continue, how do you plan to evolve the integrated method to incorporate emerging theories or frameworks that might enhance product innovation?

Reviewer #2: The authors could provide good work. However, there are some concerns to be resolved.

Would you explicitly specify the novelty of your paper? What progress against the most recent state-of-the-art similar studies was made?

The organization of the Introduction section is very unsatisfactory, and it is very messy and hard to read. Thus, this section needs rewriting in order to make it crisp and the main points of the research methodology should be mentioned clearly. This will help the readers to appreciate the novelty of the research.

You have to sharply delineate your research question and gap. Please, be more aware of deriving the question out of the relevant literature streams.

More state-of-the-art methods (models+algorithms) must be reviewed and compared to the suggested ones in this work. I suggest the authors find and read the studies performed by scholars such as Goli et al. and their groups in this regard.

Discussion of the findings was quite weak. The authors should provide a clear and insightful discussion that explains how the study presents a new perspective on the theory and practices.

There is no conceptual comparison with existing approaches, no discussion on the benefits and drawbacks of the new approach. Thus discussions and comparative analyses should be added, also it is important to compare your method with literature ones.

Please elaborate on managerial insights from the industrial cases in a distinct section. Please mention them in the abstract and conclusion sections as well.

6. PLOS authors have the option to publish the peer review history of their article (what does this mean?). If published, this will include your full peer review and any attached files.

Reviewer #1: **Yes: **Dr. A.Bovas Herbert Bejaxhin

Reviewer #2: No

---

## [Author Response · Author response to Decision Letter 0]

19 Oct 2023

Thank you very much for the questions and suggestions provided by the reviewers. We have responded to each question and suggestion in detail, and the specific content can be found in the "Response to Reviewers" document.

---

## [Decision Letter · Decision Letter 1]

27 Dec 2023

Product innovation design process combined Kano and TRIZ with AD: case study

PONE-D-23-21362R1

Dear Dr. Wei Liu,

We’re pleased to inform you that your manuscript has been judged scientifically suitable for publication and will be formally accepted for publication once it meets all outstanding technical requirements.

Kind regards,

Mazyar Ghadiri Nejad, Ph.D.

Academic Editor

PLOS ONE

Reviewers' comments:

Reviewer's Responses to Questions

**Comments to the Author**

1. If the authors have adequately addressed your comments raised in a previous round of review and you feel that this manuscript is now acceptable for publication, you may indicate that here to bypass the “Comments to the Author” section, enter your conflict of interest statement in the “Confidential to Editor” section, and submit your "Accept" recommendation.

Reviewer #2: (No Response)

Reviewer #3: All comments have been addressed

2. Is the manuscript technically sound, and do the data support the conclusions?

Reviewer #2: (No Response)

Reviewer #3: Yes

3. Has the statistical analysis been performed appropriately and rigorously? 

Reviewer #2: (No Response)

Reviewer #3: Yes

4. Have the authors made all data underlying the findings in their manuscript fully available?

Reviewer #2: (No Response)

Reviewer #3: Yes

5. Is the manuscript presented in an intelligible fashion and written in standard English?

Reviewer #2: (No Response)

Reviewer #3: Yes

6. Review Comments to the Author

Reviewer #2: Authors have improved the paper well and in my point of view it can be accepted in its current form.

Reviewer #3: Dear Author (s),

The comments were adequately addressed and I feel that this manuscript is now acceptable for publication.

7. PLOS authors have the option to publish the peer review history of their article (what does this mean?). If published, this will include your full peer review and any attached files.

Reviewer #2: No

Reviewer #3: **Yes: **Matina Ghasemi

---

## [Editor Report · Acceptance letter]

22 Jan 2024

PONE-D-23-21362R1 

PLOS ONE

Dear Dr. Liu, 

I'm pleased to inform you that your manuscript has been deemed suitable for publication in PLOS ONE. Congratulations! Your manuscript is now being handed over to our production team.

Kind regards, 

on behalf of

Assoc. Prof. Dr. Mazyar Ghadiri Nejad 

Academic Editor

PLOS ONE